# Contextualizing sacrificial dilemmas within Covid-19 for the study of moral judgment

Robin Carron[☯], Nathalie Blanc[iD]*[☯], Emmanuelle Brigaud[iD][☯]

Department of Psychology, Epsylon EA 4556, University Paul Valéry Montpellier 3, Montpellier, France

☯ These authors contributed equally to this work.
* nathalie.blanc@univ-montp3.fr

**Data Availability Statement:** the two data sets (study 1 and study 2) are uploaded as a Supporting Information file.

**Funding:** The author(s) received no specific funding for this work.

## Abstract

"Sacrificial dilemmas" are the scenarios typically used to study moral judgment and human morality. However, these dilemmas have been criticized regarding their lack of ecological validity. The COVID-19 pandemic offers a relevant context to further examine individuals' moral judgment and choice of action with more realistic sacrificial dilemmas. Using this context, the purpose of the present study is to investigate how moral responses are influenced by the contextualization of the dilemma (i.e., contextualized or not within the Covid-19 pandemic). By comparing two versions of one dilemma, Experiment 1 revealed that the more realistic version (the one contextualized within the Covid-19 pandemic) did not elicit more utilitarian responses than the less realistic version (the one not contextualized within the Covid-19 pandemic). In Experiment 2, we examined more specifically whether both the perceived realism of the dilemma and the plausibility of a utilitarian action influence moral responses. Results confirmed that the contextualization of the dilemma does not make any difference in moral responses. However, the plausibility of an action appears to exert an influence on the choice of action. Indeed, participants were more inclined to choose the utilitarian action in the plausible action versions than in the implausible action versions of the dilemma. Overall, these results shed light on the importance for future research of using mundane and dramatic realistic dilemmas displaying full information regarding a sacrificial action and its consequences.

## Introduction

Today, the lack of perceived realism of scenarios used to study moral judgment and human morality constitutes one of the main criticisms of research in this area [1]. In a well-known example, the classic trolley dilemma [2, 3], individuals are told that a runaway trolley speeding down the tracks is about to kill a group of five workers who are unable to move out of the way in time. The only way to prevent the deaths of these five is to pull a lever to redirect the trolley to another track where it will kill only one person. Is it morally acceptable to kill one person to save five others? Bauman et al. [1] argue that people's responses to this type of sacrificial dilemma have low external validity because the scenarios used are unrealistic in three distinct ways: they are low in experimental, mundane and psychological realism. According to the

**Competing interests:** The authors have declared that no competing interests exist.

definitions of Aronson et al. [4], a study is low in experimental realism when the situation is not engaging for the participants, is not impactful to them or is not taken seriously; and low in mundane realism when the situation is in no way similar to what people encounter in everyday life. In this respect, classical sacrificial dilemmas are low in experimental and mundane realism. Indeed, it seems unusual to imagine being in a situation like the one described in the trolley dilemma and to take these scenarios seriously. Moreover, in everyday life, it is rare to make choices that involve the sacrifice of human life. Finally, hypothetical dilemmas do not elicit the same psychological processes as other real-world moral situations (i.e., they are low in psychological realism).

In sum, the commonly used sacrificial dilemmas are different from more realistic moral situations and, therefore, the trade-offs between deontological principles (e.g., do not kill) and utilitarian consequences (e.g., kill one to save five) may depend on the perceived realism of the situation. In this respect, the coronavirus (Covid-19) pandemic could be viewed as an opportunity to examine moral decisions about life and death with more realistic dilemmas. In this viral pandemic context, several countries have been confronted with shortages of ventilators, intensive care beds and medical personnel. The question of the allocation of scarce medical resources was therefore raised [5]. For example, you are an emergency room physician with five dying patients who have been admitted and one patient in critical condition. Would you remove the patient from a ventilator to provide it to others? From a utilitarian point of view, approving the sacrifice of one life in order to save the lives of five others is acceptable with respect to the number of lives saved. By contrast, from a deontological point of view, it is not acceptable to sacrifice the life of one person in order to save others because the value of each human life is paramount and no one has the right to sacrifice a life, regardless of any benefits that may arise in doing so.

The two types of sacrificial dilemmas (i.e., traditional trolley dilemmas and dilemmas related to the Covid-19 pandemic) correspond to problematic situations that involve a conflict between utilitarian and deontological principles. Thus, the question is to know whether subjects' hypothetical moral responses are predictive of the moral responses they would display in real-life dilemmas (e.g., those experienced during Covid-19). In line with this question, some researchers have tried to increase the realism of sacrificial dilemmas and, consequently, their ecological validity. For instance, one solution proposed by Greene et al. [6] was to ask participants to "suspend their disbelief" while reading the scenarios and to exclude from study those who reported lack of success (i.e., those who reported being unable/unwilling to do it). In line with this, Christensen and Gomila [7] conclude their review of moral dilemma tasks by suggesting that participants be instructed that the dilemmas they are about to read are similar to those likely to occur in real-life.

## Solutions to the lack of perceived realism

**Virtual reality moral paradigms.** In an attempt to improve the realism of hypothetical dilemmas, some researchers introduced moral paradigms under Virtual Reality (VR) [8–13]. Virtual Reality allows users to be immersed in a variety of moral situations that are designed to be more realistic and believable than hypothetical versions. Because immersive VR affects human behavior as a real equivalent environmental experience [14–16], it provides an alternative for studying moral responses in a more ecological way. Note also that the realistic simulation of sacrificial dilemmas allows for the examination, in real-time, of rule-violating behavior (e.g., involving physical harm) that would be impossible to study in real world settings. Navarrete et al. [10] were the first to integrate, in immersive VR, the trolley problem consisting in switching the trolley's direction to kill one person instead of five. They found that 90% of the

participants chose the utilitarian solution, a behavior pattern that is congruent with judgments of participants from previous studies using written scenario descriptions. In another study, the participants were the drivers of the train and the same pattern of results was reported [17]. By contrast, further research found a greater endorsement of utilitarian responses in VR compared to the same scenario described textually [8, 12, 13]. Niforatos et al. [11] repeatedly exposed participants to VR and text format versions of the trolley dilemma and concluded that the VR enactment of a moral dilemma fosters utilitarian decision-making. In these studies, people become more utilitarian when they are placed in an immersive virtual environment compared to a situation in which they mentally visualize the dilemma through reading the scenario (i.e., text format). This difference could be explained by the fact that the VR format of the dilemmas could elicit artificial gaming behaviors where sacrificial action may be trivialized [8]. Another possibility that deserves to be considered is that in paper-based dilemmas, participants simply express a moral judgment, whereas in VR versions, they actually perform the action: two moral responses that are likely underlined by distinct psychological mechanisms [18]. There is a gap between what people think they would do when faced with a hypothetical sacrificial dilemma and what they would do when faced with a more realistic scenario. In line with this conclusion, Bostyn et al. [19] found that participants are more willing to be utilitarian when confronted with a real-life sacrificial dilemma. In their study, participants had to make the real-life decision to administer a very painful electroshock to a single mouse in order to save five. In a real decision, 84% of participants shocked the mouse, whereas in a hypothetical version of the same dilemma, only 66% of participants predicted that they would do that.

In the VR studies and the mouse scenario described above [19], moral dilemmas were enacted and participants experienced the feeling of "being there" (e.g., [14, 16, 20]). In these studies, they also carried out harming actions (i.e., killing or letting die) and probably experienced an illusion of reality compared to conditions in which they mentally visualized dilemmas (i.e., in which dilemmas were presented in text format). Because of their ability to provide a feeling of "being there", VR scenarios increase experimental realism [21] and thus could contribute to increasing the perceived realism of sacrificial scenarios compared to the paper-based equivalent. However, Weber et al. [22] claimed that the immersed illusion (i.e., having a sensation of being in a real place) is not sufficient to create a virtual world that is perceived as realistic; the scenario must also be close to real life. More precisely, the plausibility of the environment, addressed as "the overall credibility of the scenario being depicted in comparison with expectations" ([20], p. 3549) is a contributing factor that modulates perceived realism of virtual environments. It's probably the plausibility of scenarios that constitutes the main gap between reality and the traditional moral-dilemma paradigm. Indeed, most sacrificial dilemmas correspond to outlandish scenarios that differ considerably from moral situations that people face in real life. As Bauman et al. [1] argued, "trolley problems are unrealistic and unlike anything people encounter in the real world" (p. 545). Consistent with this criticism, Terbeck et al. [13] suggested that future VR studies might investigate moral situations that are more plausible.

**Context-related studies.**   Gold et al. [23, 24] (see also [25]) have tried to overcome the lack of realism of sacrificial dilemmas by presenting trolley problems that elicit actions and moral judgments about decisions likely to occur in real life. They used a computer animation where participants were asked to make decisions that would influence the amount of money donated to children living in an orphanage in northern Uganda. In this Orphan scenario, participants decided whether varying amounts of money, denominated in meals, would be taken away from either five children or one. For this purpose, they had the option of clicking a switch to divert a meal from one child to five others. To increase the realism of the scenario, participants were shown photos and short biographies of some of the orphans and were told that the

donation to the orphanage would depend on their decisions during the experiment. Gold et al. ([23], study 2) also used a standard trolley scenario. In this version, a lower percentage of participants said that they would take action (i.e., click the switch to save the five) than in the corresponding real-life scenario (with orphans). Note that this difference in utilitarian responses between the two scenarios is not necessarily due to perceived realism. In fact, there are two obvious differences between the two scenarios that may influence participants' responses: the type of victims (i.e., children in an orphanage vs. men working on a train track) and the harm inflicted (i.e., depriving of meals vs. killing).

As many researchers have pointed out, contextualizing the action (e.g., killing or letting die in sacrificial dilemmas) can help to study moral responses as they are made in real life (e.g., [26–28]. In Watkins and Laham's [29] research, participants responded to trolley problem scenarios in two different contexts (i.e., in a war context and a peace context). The findings demonstrated that the context changes judgment on sacrificial dilemmas: killing one person to save five was judged more acceptable in a war context than in a peace context. Following Watkins and Laham's [29] research, Christen et al. [26] wanted to understand moral judgment in more real-life trolley problem scenarios involving military, firefighting, and surveillance missions. In their study, participants made decisions about remotely piloted aircraft that resulted in sacrificial outcomes.

This type of dilemma patterned after the trolley problem scenarios is expected to be more realistic than the decontextualized ones because it is transplanted into contexts that often involve life and death decisions (i.e., in which the harmful action is more plausible). The moral decision in the trolley problem dilemma situated in a war against terror (whether or not to redirect a missile) may seem more plausible compared to the same kind of decision in the classical scenario (whether or not to redirect the trolley). However, with the trolley problem, even if a context is provided, it is difficult to imagine oneself in such a situation because the scenario described does not correspond to any lived experience. It is this lack of mundane realism that Bauman et al. [1] point out in their criticisms of sacrificial dilemmas. They argue that "trolley problems also lack mundane realism because the catastrophes depicted in sacrificial dilemmas differ considerably from the type and scale of moral situations people typically face in real life" (p. 542).

Because the COVID-19 pandemic is a real-world crisis that has confronted us with decisions about life and death, it provides a unique opportunity to address previous criticisms (the lack of mundane realism in particular) and thus to investigate moral judgment in real-world dilemmas.

## Contextualizing sacrificial dilemmas with the Covid-19 pandemic

Dilemmas posed by the Covid-19 pandemic involved trade-offs between human lives. Indeed, the massive influx of patients and shortages of healthcare resources, such as anti-virals, intensive care unit beds and mechanical ventilators, led doctors to sometimes make moral decisions about which patients to admit to the Intensive Care Unit (ICU) and for which ones to provide lifesaving resources [30, 31]. Many studies have investigated public attitudes toward ethical principles underlying these pandemic triage dilemmas (e.g., [32–34]). For some authors, sacrificial dilemmas related to the Covid-19 pandemic were perceived as an alternative to traditional sacrificial dilemmas and as a unique opportunity to overcome criticisms about the lack of realism of hypothetical moral scenarios. According to Kneer and Hannikainen [35]: « Sacrificial dilemmas related to the Covid-19 pandemic—henceforth triage/critical care dilemmas—have both experimental and mundane realism; they are real-life situations with which most participants are at least indirectly acquainted. Consequently, we suspected that their level of

psychological realism would also be high" (p.7). In one of their studies ([35], study 3), they investigated the effect of realism with dilemmas patterned after the trolley problem scenarios contextualized, or not, within the medical triage context. They observed a greater utilitarian tendency in response to triage dilemmas compared to dilemmas unrelated to Covid-19: Participants were more likely to disconnect an oxygen tank used to treat a single coronavirus patient in order to save five others than to order firefighters to stop rescuing a single person trapped in a burning house in order to save five others in another burning house. Note that this tendency to favor utilitarian response in triage dilemmas, relative to dilemmas unrelated to the Covid-19 pandemic, is not necessarily due to a difference in realism between the two types of dilemmas. Indeed, the two types of dilemmas also differed regarding the nature of the moral response: medical in the triage context and non-medical in the others.

From our perspective, the solutions provided to the lack of realism of sacrificial dilemmas are still insufficient for two reasons. First, in the vast majority of studies conducted on moral judgment, researchers did not control the perceived realism of the scenarios. They started from the assumption that the situations and actions described were perceived as realistic by the participants. Kneer and Hannikainen [35], for example, argued that sacrificial dilemmas related to the Covid-19 pandemic are high in experimental, mundane and psychological realism compared to dilemmas unrelated to Covid-19, and hypothesized that the former are perceived as more realistic than the latter. But they did not measure the perceived realism of the two types of dilemmas. Second, perceived realism is defined as a one-dimension construct. In virtual reality studies, perceived realism refers to the subjective perception of being present in the depicted scenario (i.e., experimental realism, according to Bauman et al. [1]), whereas in context-related studies, perceived realism refers to the degree to which the depicted scenario could possibly occur in the real word. Finally, in studies conducted during Covid-19, perceived realism refers to the scenario's resemblance to participants' current real-world experiences (i.e., mundane realism, according to Bauman et al. [1]). Based on this analysis of the strengths and weaknesses of previous researches conducted on moral judgment, the aim of the present study was to investigate more thoroughly the issue of realism and its influence on participants' moral responses to sacrificial dilemmas.

## Perceived realism

In a media and more specifically in a narrative environment, perceived realism is the degree to which the narrative environment reflects the real world [36, 37] and conveys coherence and genuineness [38]. Busselle and Bilandzic [38] place this construct at the center of their model of narrative comprehension. They explain that a perceived lack of realism affects the understanding of the narrative and can alter attitudes and behaviors toward the narrative. In the current state of knowledge, scholars argue that perceived realism is a multidimensional construct (e.g., [38–40]). They agree on three dimensions necessary to create a realistic narrative: plausibility, factuality and typicality (e.g., [41]). Plausibility concerns the possible occurrence of the story in reality (i.e., how likely it is that the event in the scenario could possibly happen in real life); factuality concerns actual occurrence of the story in reality (i.e., how well the event in scenario depicts something that really happened) and typicality refers to the degree to which the story is perceived to be similar to events in one's real life (i.e., how well the event in the scenario reflects people's past and present experiences) [39, 40]. Some of the previous research (the context-related studies) uses scenarios that meet certain realism requirements, such as plausibility and factuality, but do not correspond to what most people actually experience (typicality). In this regard, the Covid-19 pandemic triage dilemmas are scenarios in which all three

dimensions of perceived realism could be simultaneously included. Thus, they may provide a unique opportunity to study the role of perceived realism on moral judgment.

Because realism is a multi-faceted construct, Körner et al. [42] also raised the question of the plausibility of utilitarian actions in hypothetical sacrificial dilemmas. In the footbridge scenario, for example, the only way to save the lives of the five workers is to push a stranger off a footbridge and onto the tracks below where his body would stop the trolley. Readers of this variant of the trolley problem may find the situation plausible but the action implausible (it seems implausible that a man's body would be sufficient to stop a trolley). Körner et al. [42] showed that plausibility of action, specifically plausibility of stated consequences (i.e., how probable is it that the action will achieve the desired outcomes?) and plausibility of alternatives (i.e., how probable is it that there are other, feasible and reasonable actions to achieve the desired outcomes?), influence moral judgments. Participants judged the utilitarian action (i.e., killing one person to save five) to be more appropriate in plausible action dilemmas compared to implausible action dilemmas. Thus, the higher proportion of utilitarian responses usually reported in the trolley scenario (an impersonal dilemma) compared to the footbridge scenario (a personal dilemma) could be the result of a more plausible action in the former dilemma than in the latter. Namely, it seems more plausible to save the lives of the five workers by pulling a lever to redirect the trolley to another track than by trying to stop it by pushing a man off a footbridge and onto the tracks. In line with this interpretation, Shou et al. [43] showed that participants generally had greater certainty that the five people would survive if they decided to kill the individual in impersonal dilemmas compared to personal dilemmas: an additional result that invites us to consider the plausibility of a sacrificial action in the study of moral judgment.

In the two present studies, we had two main objectives: 1) We investigated the role of perceived realism on moral responses by distinguishing and measuring different dimensions of perceived realism in sacrificial dilemmas; 2) We attempted to address criticisms about the lack of realism of sacrificial dilemmas by comparing different versions of the triage dilemma. In study 1, we specifically investigated the perceived realism of the scenario with a version contextualized within the Covid-19 pandemic (the supposedly realistic version) and another version not contextualized within the Covid-19 pandemic (the supposedly less realistic one). We hypothesize that a scenario perceived as realistic should lead to more utilitarian responses than one perceived as less realistic. In study 2, we examined whether moral responses depend on the perceived realism of the scenario as well as on the plausibility of utilitarian actions. To that aim, we used realistic and less realistic triage dilemmas that differed in action plausibility.

## Experiment 1

### Method

**Participants.**   One hundred and fifty-five French participants took part in this study during the COVID-19 crisis, between January 9 and January 17, 2021. However, 15 of them (those who reported having a close relative or friend that had died from the virus) were excluded from analyses, leaving the present sample at $n = 140$ (126 female, $M = 29.51$, $SD = 11.31$). This exclusion criterion was determined before data collection. Indeed, to the extent that the implication of a close relative influences participants' decisions in sacrificial dilemmas (see Tassy et al., [18]), this exclusion criterion ensured that, in the context of the pandemic in which the study was conducted, the potential victim(s) of the decision had no proximity (i.e., affective or genetic) to the participant who had to make a decision. Participants were not paid for their participation.

## Materials

The TRIAGE dilemma was presented in two versions: a version contextualized within the Covid-19 pandemic (the supposedly realistic version) and another version not contextualized within the Covid-19 pandemic (the supposedly less realistic version) (Table 1). The TRIAGE dilemma presented the option of taking away an oxygen reserve or antibiotics from one person to save five others. The two versions were similar with regard to the following dimensions: the dilemma involved killing one person in order to save several others, the number of people saved was identical ($N = 5$) and the potential victims were unknown and not affectively or genetically related to the participants (according to our exclusion criterion). In addition, the utilitarian response was associated with the withdrawal of life-saving resources (i.e., an oxygen reserve or antibiotics) in the health domain which ensured that the two versions were indistinguishable in regards to ethical principles. Finally, the two versions of dilemma contained exactly the same number of words. The main difference between the two versions of the dilemma concerned the contextualization of the scenario (contextualized or not within the Covid-19 pandemic).

## Procedure and measures

Participants were tested using an online questionnaire created on the platform Qualtrics (https://www.qualtrics.com). Note that French Law does not require approval of an ethics committee when data from a survey are collected and analyzed anonymously.

After giving their informed consent, participants were randomly assigned either to the contextualized version of the TRIAGE dilemma ($N = 70$) or to the non- contextualized one ($N = 70$). The dilemma was briefly presented by stating that it refers to a serious situation that could be seen as unpleasant but requires making a difficult choice.

After reading the dilemma, participants successively provided two moral responses: a moral judgment and a choice of action. Participants were asked to be as honest as possible in their responses, knowing that there was no right or wrong answer. Immediately afterwards, they were again instructed to carefully read the same version of the dilemma presented earlier and to evaluate the dilemma from a moral perspective by answering various questions (i.e., perceived realism and plausibility of stated consequences). Finally, participants provided demographic information and were asked the following yes/no question: "have you lost a close relative or friend to COVID-19?". This question was asked at the end of the experiment and not at the beginning in order to avoid raising the saliency of the pandemic context, especially

**Table 1. Contextualized and non-contextualized versions of the TRIAGE dilemma used in experiment 1 (translated from French).**

| Versions contextualized within COVID-19 | Versions not contextualized within COVID-19 |
|---|---|
| You are the department head of a hospital in eastern France. A new coronavirus from China which causes respiratory irritation has appeared. Every day you receive more and more new patients with breathing problems. You don't have enough oxygen for all of the patients. Five new patients are admitted to the hospital's intensive care unit. Their health condition requires immediate hospitalization and the administration of oxygen for the next 15 days. There is no more oxygen available, and you have no way to get it. The only way to save the five patients is to take an oxygen tank from one of your patients who is in critical condition. If you do that, the patient will die but the other five will be saved. | You are the department head of a hospital in eastern France. A new bacterium has contaminated the water of the city. Every day you receive more and more new patients with intestinal disorders and blood poisoning. You do not have enough antibiotics for all of the patients. Five new patients are admitted to the hospital's intensive care unit. Their health condition requires immediate hospitalization and a dose of antibiotics. There are no more antibiotics available, and you have no way to get some. The only way to save the five patients is to take antibiotics from one of your patients who is in critical condition. If you do that, the patient will die but the other five will be saved. |

among participants assigned to the version of the dilemma which was not contextualized within the Covid-19 pandemic.

**Moral responses measures.**   Because there is a huge difference between what one judges as morally acceptable and what one actually does (see Tassy et al., [18]), participants responded successively to two questions, one targeting moral judgment and the other targeting choice of action. They rated to what extent the utilitarian action was appropriate or not (i.e., moral judgment task). For each version, the question was "*How appropriate is it for you to take [the oxygen / antibiotics] of one of your patients in order to save the five?*". They were also asked whether they would perform the utilitarian action (i.e., choice of action task). For each version, the question was «*Would you take [the oxygen / antibiotics] of one of your patients in order to save the five?*" These two questions were answered on a 6-point scale (1 = not at all; 6 = definitely) with higher scores being closer to utilitarian responses.

**Perceived realism measures.**   To check the effectiveness of our realism manipulation, we also assessed the perceived realism of the two versions of the TRIAGE dilemma. In line with authors who argue that realism perceptions are multidimensional (e.g., [38–40], our assessment of perceived realism included three sub-dimensions: perceived plausibility, typicality, and factuality. The question related to plausibility was "*How probable do you think it is that the event in the scenario could possibly happen in real life?*" The question related to factuality was "*How probable do you think is it that the event in the scenario depicts something that really happened?*" The question related to typicality was "*How probable do you think is it that the event in the scenario reflects people's past and present experiences?*" (see [41] for similar measures). Responses to these three perceived realism measures were all rated on a 6-point scale (1 = not at all, to 6 = definitely).

**Plausibility of stated consequences measures.**   Because the plausibility of the sacrificial action, especially the plausibility that the stated consequences will occur, might influence moral responses (see [42]), participants answered the following question "*how probable do you think is it that this action would save the five people?*" (see [42] for a similar measure). Responses were rated on the same 6-point scale (1 = not at all, to 6 = definitely).

## Results

**Manipulation check.**   A one-way analysis of variance (ANOVA) was run on three measures of perceived realism (i.e., plausibility, factuality and typicality) to evaluate the differences between the two dilemma versions (Contextualized *vs.* non contextualized within the Covid-19 pandemic).

The manipulation check confirmed that the two versions differed in perceived realism. A significant main effect was observed for plausibility $F(1, 138) = 15.94$, $p < .001$, $\eta^2_p = .10$, for factuality $F(1, 138) = 23.14$, $p < .001$, $\eta^2_p = .14$, but also for typicality $F(1, 138) = 19.61$, $p < .001$, $\eta^2_p = .12$. Participants perceived the dilemma as more realistic when it was contextualized within the Covid-19 pandemic ($M_{plausibility} = 5.30$, $SD = 1.13$; $M_{factuality} = 5.44$, $SD = 1.05$; $M_{typicality} = 5.43$, $SD = 1.11$) than when it was not contextualized ($M_{plausibility} = 4.40$, $SD = 1.51$; $M_{factuality} = 4.37$, $SD = 1.54$; $M_{typicality} = 4.43$, $SD = 1.53$).

Pearson's correlation analysis was performed to explore the correlation between the three dimensions of perceived realism. The results showed that plausibility was positively associated with factuality and typicity and that factuality was positively related to typicity, $ps < .001$ (see Table 2).

**Moral responses.**   To explore the effect of perceived realism on participants' moral responses, a repeated-measure ANOVA was conducted with the Type of dilemma (Contextualized *vs.* non contextualized within the Covid-19 pandemic) as the between-subject factor,

**Table 2. Correlations between the three dimensions of perceived realism.**

| Dimensions | 1 | 2 | 3 |
|---|---|---|---|
| 1. Plausibility | - | - | - |
| 2. Factuality | .52 | - | - |
| 3. Typicity | .65 | .67 | - |

*Note.* All ps < .001. A Pearson correlation coefficient ranging between 0.5 and 1 indicated a strong correlation.

and the Type of moral response (Judgment *vs.* Choice of action) as the within-subject factor. This analysis revealed a significant main effect of Type of moral response, indicating that responses to choice of action ($M = 3.93$, $SD = 1.50$) were overall more utilitarian than responses to judgment ($M = 3.41$, $SD = 1.54$), $F(1, 138) = 17.88$, $p < .001$, $\eta_p^2 = 0.11$. No other effects were significant.

To assess the relationship between perceived realism and moral responses, we also conducted Pearson's correlation analyses between each of the three measures of realism (plausibility, factuality and typicality) and moral responses (moral judgment and choice of action). None of these measures were significantly correlated with moral judgment or with choice of action.

In order to further test the impact of perceived realism on moral responses, we conducted two multiple regression analyses predicting judgment and choice of action separately. For each analysis, we entered the three measures of perceived realism (i.e., plausibility, factuality and typicality) as predictors. Multiple regression analysis showed that none of the perceived realism measures was a significant predictor for moral responses (all $ps > .10$, see Table 3 for a summary of regression results).

**Plausibility of stated consequences.** A one-way ANOVA was run on this measure with the Type of dilemma (Contextualized *vs.* non contextualized within the Covid-19 pandemic) as the between-subject factor. No significant difference was found between the two versions of dilemma. Overall, the probability of the action saving the five people was rated as medium ($M = 3.76$, $SD = 1.44$). Moreover, Pearson's correlation analysis showed that the plausibility of stated consequences and moral responses were positively correlated. That is, the more plausible participants rated the stated consequences, the more utilitarian their moral judgment and choice of action were ($r = .18$, $p < .05$ and $r = .22$, $p < .01$, respectively).

## Discussion

Contrary to studies dealing with the influence of realism on moral responses in sacrificial dilemmas, in this first experiment we assessed the perceived plausibility, factuality, and

**Table 3. Multiple regression analysis on judgment and choice of action with perceived plausibility, factuality and typicality of dilemma as predictors.**

| Predictors | Beta | SE (B) | t | p | Semi-partial correlation |
|---|---|---|---|---|---|
| Predicting Judgment | | | | | |
| Plausibility | 0.039 | 0.125 | 0.310 | 0.757 | 0.027 |
| Factuality | -0.061 | 0.126 | -0.485 | 0.629 | -0.041 |
| Typicity | 0.127 | 0.141 | 0.897 | 0.371 | 0.076 |
| Predicting Choice of action | | | | | |
| Plausibility | 0.120 | 0.121 | 0.997 | 0.321 | 0.085 |
| Factuality | -0.009 | 0.122 | -0.077 | 0.939 | -0.007 |
| Typicity | 0.044 | 0.137 | 0.321 | 0.749 | 0.028 |

typicality (three dimensions of perceived realism) of each employed dilemma. Although the two versions of the TRIAGE dilemma differed on these three dimensions, the more realistic version (the one contextualized within the Covid-19 pandemic) did not elicit more utilitarian responses than the less realistic version (the one not contextualized within the Covid-19 pandemic). This suggests that perceived realism did not influence moral responses. Subsequent regression analysis showed that none of the three dimensions of perceived realism was a significant predictor for moral judgment or choice of action.

One possible explanation is that participants' moral responses are not based on what they perceive as realistic but on their beliefs about the plausibility of the stated consequences. According to Körner et al. [42] (see also [43, 44]), these beliefs may influence the expected usefulness of the sacrificial action and consequently their moral decision-making. Note that the two versions of our TRIAGE dilemma were only distinguished according to perceived realism and did not differ as to the plausibility of the stated consequences, which were considered moderately plausible in both versions. Like other sacrificial dilemmas employed in moral judgment studies, our scenarios displayed low information regarding sacrificial actions and their consequences. This lack of plausibility action information in both versions of the dilemma may have led all participants to infer that the sacrificial action (i.e., killing one patient) would not with certainty produce the stated positive outcome (i.e., saving five others). Consistent with this reasoning, Kusev et al. [45] demonstrate that, in a moral dilemma, displaying full information regarding moral actions and their consequences resulted in an increase of utilitarian responses.

In Experiment 2, we examined whether both perceived realism of the scenario and plausibility of utilitarian action influence moral responses in a sacrificial dilemma. We assume that plausibility, but not necessarily dimensions of perceived realism, influences moral responses. Based on the findings of Körner et al. [42] (see also [43, 44]), but also according to our data showing that the more plausible participants rated the stated consequences, the more utilitarian their moral judgment and choice of action were, we suppose that a scenario with a plausible action should lead to more utilitarian responses than the same scenario with an implausible action, regardless of the perceived realism of the dilemma.

Furthermore, by distinguishing moral judgment and choice of action, we found that the tendency to produce utilitarian responses was higher for the choice of action than for judgment (see [18] for similar results). In experiment 2, we investigated whether the perceived realism of the scenario and the plausibility of the utilitarian action differentially affect these two distinct processes underlying moral responses.

## Experiment 2

### Method

**Participants.**    334 undergraduate students from the University of Montpellier 3, France, took part in this second study during lockdown (April 03$^{rd}$-May 03$^{rd}$, 2021). In accordance with the exclusion criterion defined before data collection (see Experiment 1 for a similar criterion), 14 participants (those who reported having a close relative or friend that had died from the virus) were excluded from the analyses. The sample size was therefore $n$ = 320 (271 female, $M$ = 20.03, $SD$ = 4.55).

### Material

We used the same TRIAGE dilemma as in Study 1. This dilemma was modified to create four versions (Table 4). The type of dilemma (contextualized or not within the Covid-19 pandemic) and the plausibility of action (plausible or implausible) varied orthogonally. Specifically, we

**Table 4. The versions of the TRIAGE dilemma added in experiment 2 (translated from French) in order to examine the plausibility of an action.**

| Versions contextualized within COVID-19 | Versions not contextualized within COVID-19 |
|---|---|
| You are the department head of a hospital in eastern France. A new coronavirus from China which causes respiratory irritation has appeared. Every day you receive more and more new patients with breathing problems. You don't have enough oxygen for all of the patients. Five new patients are admitted to the hospital's intensive care unit. Their health condition requires immediate hospitalization and the administration of oxygen for the next 15 days. There is no more oxygen available, and you have no way to get it. The only way to save the five patients is to take the oxygen tank from one of your patients who is in critical condition. | You are the department head of a hospital in eastern France. A new bacterium has contaminated the water of the city. Every day you receive more and more new patients with intestinal disorders and blood poisoning. You do not have enough antibiotics for all of the patients. Five new patients are admitted to the hospital's intensive care unit. Their health condition requires immediate hospitalization and a dose of antibiotics. There are no more antibiotics available, and you have no way to get some. The only way to save the five patients is to take antibiotics from one of your patients who is in critical condition. |
| This solution is possible because the sharing of oxygen would suffice to save them. Only this solution guarantees a 100% probability of saving them. | This solution is possible because the sharing of antibiotics would suffice to save them. Only this solution guarantees a 100% probability of saving them. |
| If you do that, the patient in critical condition will die but the other five will be saved. | If you do that, the patient in critical condition will die but the other five will be saved. |
| There is no other possible alternative. Transferring patients to another hospital is not feasible because they would not survive the transportation time. The patient in critical condition cannot share the oxygen because it would cause his or her death, and requisitioning oxygen from another hospital would take too much time. | There is no other possible alternative. Transferring patients to another hospital is not feasible because they would not survive the transportation time. The patient in critical condition cannot share the antibiotics because it would cause his or her death, and requisitioning antibiotics from another hospital would take too much time. |
| Time is short and you know that this is the one and only solution to save the five patients. | Time is short and you know that this is the one and only solution to save the five patients. |

used both the contextualized and non-contextualized versions of the dilemma described in experiment 1. Each of these two scenarios were modified to create a plausible action version and an implausible action version. In order to manipulate the plausibility of action, we referred to the methodology used by Körner et al. [42] in which two aspects of plausibility were manipulated: the plausibility of the stated consequences and the plausibility of alternatives (two aspects influencing moral responses in an independent and additive manner). The two aspects were combined; that is, in the plausible action versions, the consequences of the utilitarian option were certain (i.e., this solution guarantees a 100% probability of saving the five) and the utilitarian option was the only way to save people (i.e., there were no other possible alternatives). In the implausible action versions of the dilemma, the consequences of the utilitarian option were not certain and better alternative actions could be imagined. In summary, we had four versions of the TRIAGE dilemma: two versions contextualized within the COVID-19 pandemic with either a plausible or implausible action; and two versions not contextualized within the COVID-19 pandemic with either a plausible or implausible action. Finally, both the plausible and implausible action versions of the dilemma contained exactly the same number of words.

## Procedure and measures

The procedure and measures are similar to those used in experiment 1. Participants were tested using an online questionnaire created on the platform Qualtrics (https://www.qualtrics.com). They were informed that their responses would remain anonymous according to the Data Protection law.

After giving their informed consent, participants were randomly assigned to one of the four versions of the TRIAGE dilemma. First, they read the dilemma and successively provided two

moral responses: a moral judgment and a choice of action. Second, we measured the perceived realism of the dilemma and the plausibility of the action for each version. In this second part, participants carefully reread the same version of the dilemma and evaluated it by answering the same questions as in experiment 1 (i.e., three questions related to perceived realism and one question related to the plausibility of the stated consequences). In addition, based on the findings of Körner et al. [42] and in order to check the effectiveness of our plausibility manipulation, we also assessed the plausibility of better alternative actions with the following question: *"How plausible is it that there are no better alternative actions—no reasonable actions to [achieve outcome = save the five patients]?"*. Then, participants had to rate it on the same 6-point scale as the other questions. Finally, participants provided demographic information and were asked "have you lost a close relative or friend to COVID-19?" (yes/no question). As in Experiment 1, we asked this question at the end of the experiment in order to avoid raising the saliency of the pandemic context, especially among participants assigned to the version of the dilemma which was not contextualized within the Covid-19 pandemic.

## Results

**Manipulation check.** To check the effectiveness of our realism manipulation, we conducted separately 2 X 2 analyses of variance (ANOVAs) with the Type of dilemma (Contextualized vs. non contextualized within the Covid-19 pandemic) and the Plausibility of action (plausible or implausible) as between-subject factors, on the three measures of perceived realism (i.e., plausibility, factuality and typicality).

As in Experiment 1, we observed a significant main effect of the type of dilemma on each of the three dimensions of perceived realism indicating that the contextualized versions within Covid-19 were judged more realistic than the non-contextualized ones. Specifically, participants perceived the dilemma as more plausible, more factual and more typical ($M_{plausibility}$ = 5.15, $SD$ = 1.38; $M_{factuality}$ = 5.27, $SD$ = 1.09; $M_{typicality}$ = 5.65, $SD$ = 0.70) when it was contextualized within the Covid-19 pandemic than when it was not contextualized ($M_{plausibility}$ = 4.50, $SD$ = 1.45; $M_{factuality}$ = 4.90, $SD$ = 1.18; $M_{typicality}$ = 5.04, $SD$ = 1.11), $ps$ < .01. It should be noted that the perceived realism of the scenario was not influenced by the plausibility of the action, regardless of the type of dilemma (Contextualized vs. non-contextualized within the Covid-19 pandemic). In short, there was no main effect of Plausibility of action, nor significant interaction with Type of dilemma on the three dimensions of perceived realism (i.e., all versions of the dilemma were judged equally plausible, factual, and typical). Once again, Pearson's correlation analysis showed that these three dimensions of perceived realism were positively and significantly correlated, $ps$ < .001.

Judgments of plausibility of action (i.e., plausibility of stated consequences and plausibility of better alternative actions) were also entered separately into 2 (Type of dilemma: Contextualized vs. non contextualized within the Covid-19 pandemic) X 2 (Plausibility of action: plausible vs. implausible) between-subjects ANOVAs. These analyses showed a significant main effect of the plausibility of an action, indicating that the stated consequences were rated as more certain in the plausible action versions ($M$ = 4.87, $SD$ = 1.51) than in the implausible action versions of the dilemma ($M$ = 3.56, $SD$ = 1.65), $F(1,316)$ = 54.43, $p$ < .001, $\eta_p^2$ = 0.15. In addition, participants judged it more likely that there were no better alternative actions in the plausible action versions ($M$ = 3.12, $SD$ = 1.61) than in the implausible action versions of the dilemma ($M$ = 2.20, $SD$ = 1.43), $F(1, 316)$ = 29.54, $p$ < .001, $\eta_p^2$ = 0.09. Regarding the manipulation check, no other effects were significant.

**Moral responses.** To explore the effect of both perceived realism and plausibility of action on individuals' moral responses, a repeated-measure ANOVA was conducted with the Type of

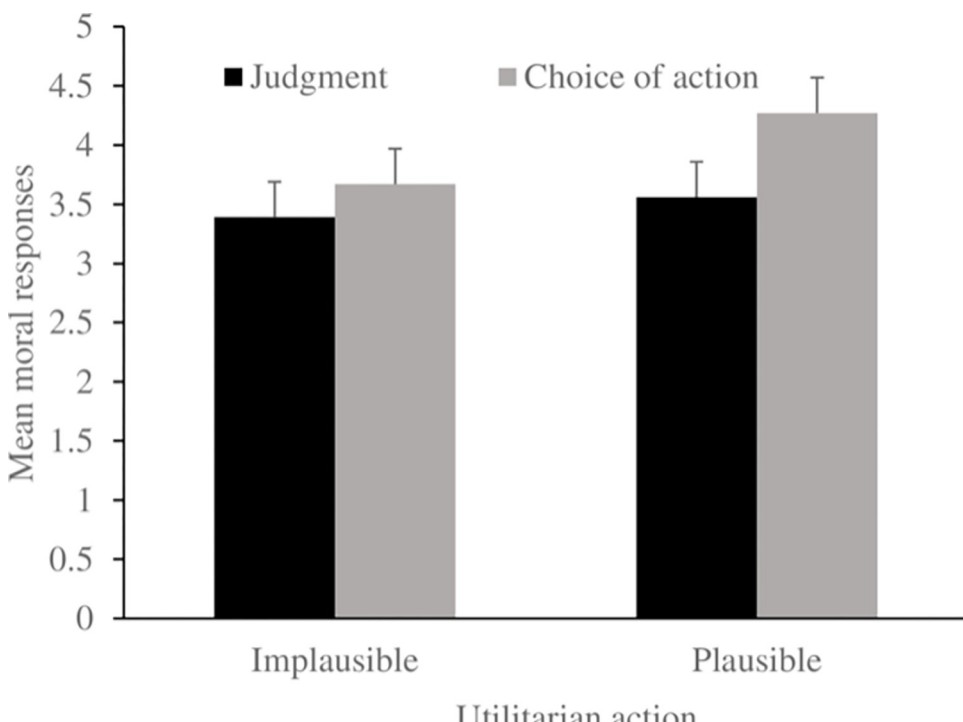

**Fig 1. Mean moral responses as a function of type of moral response (judgment vs. choice of action) and plausibility of sacrificial action (plausible vs. implausible).** Higher scores (max = 6) are closer to utilitarian responses.

dilemma (Contextualized vs. non contextualized within the Covid-19 pandemic) and the Plausibility of action (plausible or implausible) as between-subject factors, and the Type of moral response (Judgment *vs.* Choice of action) as within-subject factor. The Type of moral response main effect was significant, $F(1,316) = 32.33$, $p < .001$, $\eta_p^2 = 0.09$. As in Experiment 1, responses for choice of action ($M = 3.97$, $SD = 1.46$) were overall more utilitarian than responses to judgment ($M = 3.47$, $SD = 1.45$). The Plausibility of action main effect was also significant, $F(1,316) = 7.69$, $p < .01$, $\eta_p^2 = 0.02$, indicating that moral responses were more utilitarian in the plausible action versions ($M = 3.91$, $SD = 1.20$) than in the implausible action versions of the dilemma ($M = 3.53$, $SD = 1.22$). A significant Plausibility of action x Type of moral response interaction showed that participants tended to be more utilitarian for the plausible action versions compared to the implausible action versions of the dilemma in their choice of action only (Fig 1), $F(1,316) = 6.13$, $p = .01$, $\eta_p^2 = 0.02$. Post hoc analysis (Scheffé test) revealed that participants were more inclined to choose the utilitarian action in plausible action versions than in implausible action versions of the dilemma, $p = .004$. Their moral judgment did not differ according to the plausibility of action, $p = .80$. In addition, in plausible action versions of the dilemma, they appeared to be more utilitarian in their choice of action than in their judgment, $p < .001$. No other effects were significant.

To further test the impact of perceived realism and plausibility of action on moral responses, we conducted two multiple linear regression analyses. For each regression calculation, we entered the three measures of perceived realism (plausibility, factuality and typicity) and the two measures of plausibility of action (Plausibility of stated consequences and Plausibility of better alternative actions) as predictors. As in Experiment 1, results showed that none of the perceived realism measures was a significant predictor of moral responses: perceived

**Table 5. Results of linear regression analyses predicting moral choice ratings from ratings of perceived realism and plausibility of action.**

| Predictors | Beta | SE (B) | t | p | Semi-partial correlation |
|---|---|---|---|---|---|
| Predicting Judgment | | | | | |
| Plausibility | 0.124 | 0.069 | 1.782 | 0.076 | 0.100 |
| Typicity | 0.060 | 0.110 | 0.544 | 0.587 | 0.031 |
| Factuality | -0.086 | 0.086 | -0.994 | 0.321 | -0.056 |
| Plausibility of stated consequences | 0.066 | 0.052 | 1.249 | 0.213 | 0.070 |
| Plausibility of better alternative actions | 0.051 | 0.056 | 0.907 | 0.365 | 0.051 |
| Predicting Choice of action | | | | | |
| Plausibility | -0.038 | 0.068 | -0.555 | 0.579 | -0.031 |
| Typicity | 0.125 | 0.108 | 1.154 | 0.249 | 0.065 |
| Factuality | -0.034 | 0.084 | -0.397 | 0.692 | -0.022 |
| Plausibility of stated consequences | 0.190 | 0.051 | 3.697 | < .001 | 0.204 |
| Plausibility of better alternative actions | 0.111 | 0.055 | 2.032 | 0.043 | 0.114 |

plausibility, factuality and typicity of the dilemma did not predict judgment or choice of action (all $ps > .10$, see Table 5 for a summary of regression results). Interestingly, both perceived components of the plausibility of an action (i.e., plausibility of stated consequences and plausibility of better alternative actions) predicted individuals' choice of action. Indeed, a significant regression equation was found, $F(5,314) = 6.07$, $p < .001$), with an $R^2$ of .09: The more participants judged the consequences of the utilitarian action as certain and rejected the possibility of better alternatives, the more they endorsed the utilitarian action. Note that the plausibility of the stated consequences predicted choice of action better than the plausibility of alternatives. Neither of these two measures was a significant predictor of moral judgment.

## Discussion

In this second experiment, we examined whether both the perceived realism of the scenario and the plausibility of the utilitarian action influence moral responses in a sacrificial dilemma. In investigating perceived realism, we again distinguished different dimensions of perceived dilemma realism (i.e., plausibility, factuality and typicity). As in Experiment 1, we found that none of these dimensions of perceived realism was a significant predictor for moral judgment or choice of action. As expected, plausibility of action influenced moral responses: participants were more inclined to choose the utilitarian action in plausible action versions than in implausible action versions of the TRIAGE dilemma. It is noteworthy that this effect was not observed for moral judgments but only for choice of action. This result confirms and extends previous findings showing that plausibility of action, especially plausibility of stated consequences and plausibility of better alternative actions, influences moral responses in sacrificial dilemmas. Initial evidence for this effect came from Körner et al. [42], (see also [43, 44]), who showed that participants endorsed the utilitarian killing response if the stated consequences seemed certain and if it seemed plausible that there were no better alternatives to save the lives of five people. Furthermore, the fact that plausibility of action did not have a significant influence on participants' moral judgment sheds light on the necessity of distinguishing judgment and choice of action in moral responses.

According to Tassy et al. [18] (see also [46]), moral judgment and moral choice may be underlied by distinct psychological mechanisms. In our studies, participants were asked to make a decision related to the allocation of healthcare resources. In the context of a severe pandemic, such as Covid-19, we know that maximizing survival rates is the most important aim (e.g., [5, 30, 47]). Therefore, it can be assumed that, faced with a TRIAL dilemma, participants

were able to distinguish between what is morally acceptable (i.e., judgment) and what decisions must be made to manage the crisis (i.e., choice of action). Thus, they might judge the sacrificial action mainly according to moral rules ("it's forbidden to kill", "do no harm"). On the other hand, they might choose to act or not to act mainly according to the outcomes they could reasonably expect (i.e., "saving a maximum number of lives"). In other words, unlike moral judgment, the choice of action could result mainly from controlled cognitive processes, as identified in the dual-process theory [48–50]. In line with this theory, primitive emotional responses that prime us not to endorse harmful actions in the name of moral rules can be overridden by an effortful deliberate processing of outcomes that drive rational choices, seeking to maximize the wellbeing of a larger number of people. If people consider plausible the suggestion that killing one person will guarantee positive consequences (i.e., saving five others), they should rationally choose the sacrificial action. In line with this explanation, our results reveal that if taking *[the oxygen / antibiotics] of one patient* saves with certainty the five and if there is no other way (via alternative actions) to save them, people tend to be more utilitarian in their choice of action than when the consequences are uncertain and alternative actions exist. When faced with a scenario with a plausible action, they also tend to be more utilitarian in their choice of action than in their judgment. However, the plausibility of an action does not predict their moral judgment.

To explain moral responses, the rational perspective might not be sufficient to account for quick decision-making in real-world situations, such as the triage dilemma. Indeed, it could be the case that, in these mundane as well as dramatic realistic scenarios in which urgent decisions must be made, the choice of an action is driven by simple and fast intuitive processes. According to one intuitionist perspective, the Agent Deed Consequence (ADC) model [51, 52], people use and evaluate three kinds of intuitions while making moral decisions: the person who is doing something, specifically with respect to his traits or intention (the Agent-component, A), the deed or what is being done (the Deed-component, D), and the consequences or outcomes that resulted from the deed (the Consequences-component, C). The positive or negative intuitive evaluations of each of these components can be used simultaneously in a comparative framework in order to produce a positive or negative judgment of moral acceptability. Therefore, a wrong deed (e.g., killing or harming a person) may be more acceptable if the agent has good intentions and the action has a good consequence (e.g., saving five people). According to Dubljević [53] (see also [54]), this idea that different components of moral intuitions (A, D, and C) simultaneously drive moral responses is helpful to explain the flexible but stable nature of moral judgment. In our Experiment 2, the negative evaluation of the sacrificial action (taking [the oxygen / antibiotics] of one patient) could be counterbalanced by the positive evaluation of the consequences (being certain to save the five). Consistent with the predictions of the ADC model, our results suggest that the consequences or outcomes that resulted from the deed (the D component) may be less relevant in moral judgment than in moral choice.

Such a promising explanation should be considered with caution, and further research is needed to investigate the processes and mechanisms underlying judgment and choice of action in sacrificial moral dilemmas. These processes and mechanisms might differ depending on whether individuals are faced with real-world dilemmas or hypothetical dilemmas.

## Limitations

Several limitations, inherent to the nature of the sacrificial dilemma used in our studies, should be noted.

First, although the TRIAGE dilemma in health care is not a new topic (similar dilemmas have arisen in organ transplantations, for example), the version based on oxygen allocation

was considered more realistic than the one based on antibiotic allocation because France and other counties were actually faced with a critical shortage of the lifesaving machines during the Covid-19 pandemic. Outside of this pandemic context caused by a respiratory virus, our dilemma would have no ecological validity.

Second, even if our two versions of the dilemma (i.e., the contextualized one and the non-contextualized one) were strictly comparable and differed only in perceived realism, the lack of difference in moral responses between the two versions should be interpreted with caution. It does not necessarily mean that perceived realism does not influence the moral response. Indeed, in both versions, the moral response was a medical decision that may have interfered with the effect of perceived realism in a highly salient pandemic context at the time of the experiment. In this context, the moral rule ("it's forbidden to kill") can be outweighed by moral considerations such as the harm that will be prevented if these rules are set aside. Kahane [55], referring directly to medical triage, argues that "overruling a moral rule in emergency context when lives are at stake is part of commonsense morality" (p.556). In this sense, the *duty of rescue* may have functioned as a prescriptive (or injunctive) norm guiding participants' judgments and choice of action. Further research should investigate the role of perceived realism on moral responses in other dramatic situations involving real life and death decisions.

Third, because triage issues in a pandemic context reflect a given society's moral standards, the moral responses collected in the present studies are not necessarily the same in every country around the world.

## Conclusion

Despite their limitations, the present studies are the first to investigate the role of perceived realism on moral response by measuring the realism of proposed scenarios. Contrary to previous studies, we compared different versions of the same sacrificial scenario (a triage dilemma contextualized or not within the Covid-19 pandemic) by controlling for the perceived realism of each version. Although the contextualized version was perceived as more realistic than the non-contextualized version, the moral responses were the same. Thus, our results do not allow us to conclude that there is an effect of perceived realism on moral responses but highlight one key factor: the plausibility of the sacrificial action. In our studies, regardless of the degree of realism, participants chose the utilitarian action if the stated consequences seemed certain and if it seemed plausible that there were no better alternatives to save the lives of five people. This result supports the findings of several authors (e.g., [42–45]) who had already highlighted the influence of the plausibility of sacrificial action on moral responses.

It is also worthwhile to highlight that distinguishing between the perceived realism of the scenario and the plausibility of the action leads to a reconsideration of criticisms made concerning hypothetical dilemmas: instead of focusing on the lack of realism (a factor that does not seem to influence moral responses), the certainty of action (a determinant of moral choices) deserves to be more thoroughly considered in moral dilemma studies. This is closely linked to an idea mentioned by Dubljević and Racine [51]: "In the case of hypothetical dilemmas, uncertainty related to the response options is silenced" (p.12). In that case, participants may therefore infer probabilities for themselves concerning certainty of outcomes. To solve this problem (i.e., subjective ratings of certainty), Shou et al. [43] suggested controlling or measuring subjective judgments regarding dilemma outcomes.

In addition, Bauman et al. [1] suggested to study moral responses in a more ecological way by creating "new scenarios involving the same types of trade-offs as sacrificial dilemmas, but presenting them in a way that is more consistent with how people might face these trade-offs

in the real world" (p.546). As an example, Bauman et al. [1] were already referring to medical professionals who must make decisions about the allocation of scarce medical resources. In line with this recommendation, we suggest that future research should use both mundane and dramatic realistic dilemmas displaying full information regarding the three components needed to make moral decisions: the agent, the deed in itself and the consequences of the deed (Dubljević et al., [54]).

## Supporting information

**S1 Data.**
(XLSX)

## Author Contributions

**Conceptualization:** Robin Carron, Nathalie Blanc, Emmanuelle Brigaud.

**Data curation:** Emmanuelle Brigaud.

**Investigation:** Robin Carron, Nathalie Blanc, Emmanuelle Brigaud.

**Methodology:** Robin Carron, Nathalie Blanc, Emmanuelle Brigaud.

**Supervision:** Nathalie Blanc, Emmanuelle Brigaud.

**Writing – original draft:** Robin Carron, Nathalie Blanc, Emmanuelle Brigaud.

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
