## [Decision Letter · Decision Letter 0]

18 Mar 2022

PONE-D-22-03571Contextualizing sacrificial dilemmas within COVID-19 for the study of moral judgmentPLOS ONE

Dear Dr. Blanc,

Thank you for submitting your manuscript to PLOS ONE. After careful consideration, we feel that it has merit but does not fully meet PLOS ONE’s publication criteria as it currently stands. Therefore, we invite you to submit a revised version of the manuscript that addresses the points raised during the review process.

This manuscript has potential. Yet, especially reviewer one has major concerns that have to be taken in consideration and properly addressed before I can take a decision on publication. Also the point raised by the second reviewer is not minor. The request is for very significant revisions. 

We look forward to receiving your revised manuscript.

Kind regards,

Sara Rubinelli

Academic Editor

PLOS ONE

Journal Requirements:

Reviewers' comments:

Reviewer's Responses to Questions

**Comments to the Author**

1. Is the manuscript technically sound, and do the data support the conclusions?

Reviewer #1: Partly

Reviewer #2: Yes

2. Has the statistical analysis been performed appropriately and rigorously? 

Reviewer #1: I Don't Know

Reviewer #2: I Don't Know

3. Have the authors made all data underlying the findings in their manuscript fully available?

Reviewer #1: Yes

Reviewer #2: No

4. Is the manuscript presented in an intelligible fashion and written in standard English?

Reviewer #1: Yes

Reviewer #2: Yes

5. Review Comments to the Author

Reviewer #1: The experiments conducted are very interesting but I have some doubts about their methodology and purported realism, which is very important given that that’s the topic under investigation.

Lengthy introduction gives useful introduction to the literature in this area. However, there are a few shortcomings, mainly from the ethical perspective:

The language used here presents utilitarianism as unethical/an opponent of ethics. “From an ethical point of view” should say “from a deontological” point of view, or some other moral system. “From a utilitarian point of view, approving the sacrifice of one life in order to save the lives of five others is morally acceptable with respect to the number of lives saved. By contrast, from an ethical point of view, it is not acceptable to sacrifice the life of one person in order to save others because the value of each human life is paramount and no one has the right to sacrifice a life.”

It's not the “footbridge dilemma”, it’s the most famous variation of the trolley problem. And “Someone is standing on the tracks of a bridge” doesn’t make sense.

Why use the fat man example rather than the classic trolley problem? Using the fat man example instead makes it sound like you intended people to die when you deny them an intensive care bed, whereas that’s not the case.

More argumentation is needed to back up the claim that “emotion makes utilitarian decisions harder to make”. For many people the utilitarian choice is justified by an appeal to emotion.

Also, people might intuitively believe that they should act deontologically, but later on reflection regret not making the utilitarian decision.

It’s true that the footbridge dilemma is unrealistic. However the main trolley problem is much more realistic and that’s another reason it should be used in this paper instead.

Again, the “switch dilemma” isn’t “a version of the trolley problem”, it IS the trolley problem.

One drawback of the VR approach is that it risks making the experiment seem more like a game than an abstrac scenario.

I’m not the greatest fan of trolley problems, but it’s overstating it to say ““trolley problems are unrealistic and unlike anything people encounter in the real world”. Anyone who has swerved the car to avoid a squirrel or fox, putting their family at risk, has experienced a real life trolley problem. And as you argue in the paper, doctors on the ICU face similar issues.

Doctors have always been making ‘moral decisions about which patients to admit to the Intensive Care Unit (ICU) and for which ones to provide lifesaving resources” regardless of the pandemic.

ICU sacrifical dilemmas might seem more realistic than trolley problems, but they’re often greatly oversimplified in a way that makes them clinically unrealistic, even if they seem real to members of the public.

The footbridge dilemmas is also implausibly because it seems unlikely the decision-maker could life/shove such a heavy person, and also because there would be no way of knowing whether he would stop the train or not. In contrast, the main trolley problem is much more plausible as it lacks these “stupid” features.

Why were those who had lost a relative to Covid excluded after participation rather than before, if the criterion was decided before collection? That means that their time was wasted. An ethics committee would have picked up on this if the study had been reviewed.

The triage dilemma used in study 1 is unrealistic, at least compared to the ICU bed example. Hardly any hospitals ran out of oxygen and none that I’m aware of ran out of antibiotics. Given the emphasis on plausibility this is unfortunate. The oxygen example also differs from antibiotics (and the ICU bed) because it’s normaly certain that a patient will die without oxygen but that’s not the case for the other two. In addition, the antibiotics example seems implausible because five patients would need five times as much antibiotics as one patient, and antibiotics courses need to be taken for days. To me the oxygen scenario is less plausible than the ICU bed one, but the antibiotic one is deeply implausible which is a major issue given that it was used only in the example that’s supposed to be less plausible for the reason that it’s decontextualised. This highlights that it seems to be a methodological flaw to have oxygen in the contextualised example, but antibiotics in the noncontextualised one? That introduces an unnecessary variable that could act as a confounder. Also, looking at the examples it’s clear that they’re both contextualised; it’s just that one is a fictional and one a real-world context.

“Participants were asked to be as honest as possible in their 335 responses, knowing that there was no right or wrong answer.” Were participants told that there is no right or wrong answer in ethics? That’s a very contentious morally relativistic statement that many ethicists would disagree with.

The results section is very short and could use more textual description of the results.

Also, it would probably make sense to describe the Methods of both studies in one section, the Results in another, and then have a combined discussion.

It seems rather problematic to characterise a scenario where the outcome is uncertain as less plausible; in real life the outcome is often uncertain, ie, the more uncertain the more realistic it is. Yet the authors say “In the implausible versions of the dilemma, the consequences of the utilitarian option were not certain”; to me, that sounds very plausible.

Why does the “plausible” version of the Covid-19 scenario in study 2 say “a virus

causing respiratory irritation” rather than stating that it’s Covid-19?

The only difference between the plausible and implausible versions is the added explanation at the end. To many people that part would be obvious, so it’s not clear how it can be used to differentiate based on plausibility.

Also, it’s extremely implausible to say that “this solution guarantees a 100% probability of saving them” – there’s never a 100% probability of saving any patient, much less one who needs oxygen on the intensive care unit, where around half of patients die.

The same problem applies to the plausible version of the antibiotics example – it remains extremely implausible and the explanation that’s designed to increase plausibilty actually descreases it by making implausible statements. However, the antibiotic example because, while denying oxygen will almost certainly kill a patient, providing antibiotics only increases chances of survival.

Telling participants that one option guarantees 100% that five lives will be saved clearly reduces real-world plausibility, but increases certainty and makes the choice easier.

Given all these issues, I’m not sure how valid any of the results are.

“On the other hand, they might choose to act or not to act mainly according to the outcomes they could reasonably expect (i.e., “saving a maximum 618 number of lives”), rather than following their moral considerations”

Once again, this implies that utilitarian principles don’t count as moral considerations. A utilitarian would regarded maximising lives saved as the moral and ethical thing to do. The fact that a deontologist would disagree doesn’t mean that the utilitarian is wrong or unethical.

“Our results reveal that if taking [the oxygen / antibiotics] of one patient saves with certainty the five and if there is no other way (via alternative actions) to save them, people tend to be more utilitarian in their choice of action than when the consequences are uncertain and alternative actions exist.”

Of course they do – but certainty is deeply implausible, which is problematic for this study,.

Reviewer #2: These authors have conducted two experiments to explore factors that may affect hypothetical moral responses and actions. With regard to their finding that the respondents view the COVID-19 scenario as more realistic than the antibiotic scenario, it seems quite clear that the difference is attributable to the current pandemic. From my perspective as a clinician ethicist, those two scenarios are ethically indistinguishable. It is interesting that people tended to say that they would be more likely to act than to say that action was morally appropriate. The impact of certainty of the ability of action to save more people makes sense and adds support to existing literature. I found the authors' decision to divide the methods, results, discussion into two parts to be quite odd. They should unite them into one.

I am confused about one thing -- you said that consent is not required but you obtained it from the students. that seems desirable but inconsistent.

6. PLOS authors have the option to publish the peer review history of their article (what does this mean?). If published, this will include your full peer review and any attached files.

Reviewer #1: No

Reviewer #2: No

---

## [Author Response · Author response to Decision Letter 0]

4 May 2022

Please see the file entitled "response reviewers" we upload with the revised version of the manuscript. Regarding the data set linked to the results presented in the manuscript, of course we agree to share these data (Please see the uploaded file).

---

## [Decision Letter · Decision Letter 1]

31 May 2022

PONE-D-22-03571R1Contextualizing sacrificial dilemmas within COVID-19 for the study of moral judgmentPLOS ONE

Dear Dr. Blanc,

Thank you for submitting your manuscript to PLOS ONE. After careful consideration, we feel that it has merit but does not fully meet PLOS ONE’s publication criteria as it currently stands. Therefore, we invite you to submit a revised version of the manuscript that addresses the points raised during the review process.

One of the reviewers is still not satisfied with the revision and makes important comments on how to further revise the paper. Please also check the English professionally, as it should be improved. The introduction also needs to be shortened and made more compelling. 

We look forward to receiving your revised manuscript.

Kind regards,

Sara Rubinelli

Academic Editor

PLOS ONE

Reviewers' comments:

Reviewer's Responses to Questions

**Comments to the Author**

1. If the authors have adequately addressed your comments raised in a previous round of review and you feel that this manuscript is now acceptable for publication, you may indicate that here to bypass the “Comments to the Author” section, enter your conflict of interest statement in the “Confidential to Editor” section, and submit your "Accept" recommendation.

Reviewer #1: All comments have been addressed

Reviewer #2: (No Response)

2. Is the manuscript technically sound, and do the data support the conclusions?

Reviewer #1: (No Response)

Reviewer #2: Partly

3. Has the statistical analysis been performed appropriately and rigorously? 

Reviewer #1: (No Response)

Reviewer #2: I Don't Know

4. Have the authors made all data underlying the findings in their manuscript fully available?

Reviewer #1: (No Response)

Reviewer #2: Yes

5. Is the manuscript presented in an intelligible fashion and written in standard English?

Reviewer #1: (No Response)

Reviewer #2: Yes

6. Review Comments to the Author

Reviewer #1: (No Response)

Reviewer #2: Ultimately, I am not sure whether this adds significantly to the literature. I understand the difference between judgment and choice of action, but the problem is that the choice of action in this setting is only hypothetical. What one should care about is what people actually do when confronted with the dilemma. It is helpful that the authors assessed the perceived realism of their scenarios, but distressing that the scenarios made so little difference. Rereading this, I have further methodologic concerns. They need to say, first, what their recruitment and exclusion criteria were (and justify the latter, which was not done; it may not have been justifiable to exclude them only after they had already been asked to revisit their loss) and then they need to say what their analytical strategy was -- as it is, it reads almost like data mining (I hate to say it). And what I wanted you to do was present Experiment1 and then Experiment 2 and then discuss them together -- as it was, there was a good bit of redundancy. Moreover, unlike Reviewer 1, I thought that the introductory literature review was way too long.

7. PLOS authors have the option to publish the peer review history of their article (what does this mean?). If published, this will include your full peer review and any attached files.

Reviewer #1: No

Reviewer #2: No

---

## [Author Response · Author response to Decision Letter 1]

6 Jul 2022

Reviewer’s comments

I have re-checked this paper and I would advice to shorten the introduction by clearly introducing the topic, why it is relevant, what is the state of the art in the field and why this manuscript addresses that topic and with which purpose/objective. 

Currently, the introduction is 13 pages and the reader gets lost about the focus, its relevance and ground in the literature. 

Thank you for your comments that helped us to revise the manuscript. We agree that the introduction section was too long. As requested, we shortened and reorganized the introduction so that the way the manuscript now addresses the topic is made more salient. Three pages were eliminated from the introduction section. 

Following your recommendations, we began by directly introducing the topic with the major criticism of dilemmas classically used to study moral judgment: their lack of realism (Bauman et al., 2014). This criticism calls into question the ecological validity of measured moral responses. In line with previous research that has attempted to address this criticism, the goal of the two studies presented in this article is to 1) propose more realistic sacrificial dilemmas (the triage dilemma by using the crisis context of Covid-19), but also to measure the realism of the scenarios used. Indeed, even if the literature proposes several solutions to improve this realism (solutions that we present and criticize in our state of the art), to our knowledge no study measures the perceived realism of the dilemmas used. It is thus impossible to determine the role of this factor on moral responses. 

The contribution of our work is therefore to study moral responses in sacrificial dilemmas which are designed to be more realistic than those used so far, and also to control the perceived realism of the dilemmas studied in order to be able to draw scientific conclusions regarding the role of this factor on moral responses. 

To this end, we propose a sequence of two experimental studies in which perceived realism is assessed in its different aspects: the realism of the scenario (i.e., its plausibility, factuality, and typicity) in Study 1, and the realism of the scenario associated with the plausibility of the sacrificial action in Study 2. 

Finally, these studies contribute to shedding light on the question of the perceived realism of the dilemmas on two quite different aspects of moral response (Tassy et al., 2013): the judgment we make about a moral response (i.e., "is it morally acceptable?") and the intended action (i.e., the choice we would make if faced with a sacrificial dilemma). To the extent that, for obvious reasons, our participants could not actually act (i.e., could not sacrifice one person or let 5 persons die), measuring their intended action when faced with a sacrificial dilemma (contextualized within the Covid-19 pandemic or not) sheds light on the influence of perceived realism on choice of action.

Reviewer’s comments

For the other comments, I would better explain the methodology of the study and everything that can help understanding the results. 

We agree that the elements provided in the manuscript relating to the exclusion criterion were insufficient. To remove any ambiguity on this subject, we added information in the method section:

Before conducting the study, it was clear to us that participants with a relative or close friend who had died of Covid-19 should be excluded. This exclusion criterion is based on Tassy et al. (2013)’s results: they showed that emotional or genetic proximity between participants and dilemma protagonists influences moral responses. Indeed, responses (and especially the choice of action) are less utilitarian when the potential victim (i.e., the one to be sacrificed) is a relative (e.g., a brother, a cousin, a friend) than when he/she is a stranger. Given the number of victims of the Covid-19 pandemic, the likelihood that participants had a relative or close friend who died of Covid-19 had to be considered. Of the 489 participants, 29 were excluded on the basis of this criterion. This allowed us to avoid any bias coming from closeness of relationship of the decision-maker with the potential victim, a factor determining moral responses. 

We decided not to ask the question “have you lost a close relative or friend to COVID-19” at the beginning of the experiment for the following reason: Asking the question at the beginning would have raised the saliency of the Covid-19 pandemic context. It was indeed risky to ask this question at the beginning of the experiment for an obvious reason: in the condition with the non-contextualized version of the TRIAGE dilemma, participants were not expected to make the connection with the Covid-19 pandemic. In other words, raising the saliency of the covid-19 pandemic context could have interfered with the context variable we manipulated in this study (dilemmas contextualized and non-contextualized within Covid-19) and introduced a bias.

Of the 489 participants, 29 participants (15 in Study 1 and 14 in Study 2) were excluded on the basis of this criterion. All participants were volunteers and those who were finally excluded from the sample were not informed that their contribution to this study was not taken into account due to this exclusion criterion.

Reviewer’s comments

As for the English, once the paper has been revised I suggest rechecking the wording. 

A native English speaker rechecked the wording.

---

## [Decision Letter · Decision Letter 2]

10 Aug 2022

Contextualizing sacrificial dilemmas within COVID-19 for the study of moral judgment

PONE-D-22-03571R2

Dear Dr. Blanc,

We’re pleased to inform you that your manuscript has been judged scientifically suitable for publication and will be formally accepted for publication once it meets all outstanding technical requirements.

Kind regards,

Sara Rubinelli

Academic Editor

PLOS ONE

Additional Editor Comments (optional):

Reviewers' comments:

Reviewer's Responses to Questions

**Comments to the Author**

1. If the authors have adequately addressed your comments raised in a previous round of review and you feel that this manuscript is now acceptable for publication, you may indicate that here to bypass the “Comments to the Author” section, enter your conflict of interest statement in the “Confidential to Editor” section, and submit your "Accept" recommendation.

Reviewer #2: (No Response)

2. Is the manuscript technically sound, and do the data support the conclusions?

Reviewer #2: Yes

3. Has the statistical analysis been performed appropriately and rigorously? 

Reviewer #2: I Don't Know

4. Have the authors made all data underlying the findings in their manuscript fully available?

Reviewer #2: Yes

5. Is the manuscript presented in an intelligible fashion and written in standard English?

Reviewer #2: Yes

6. Review Comments to the Author

Reviewer #2: (No Response)

7. PLOS authors have the option to publish the peer review history of their article (what does this mean?). If published, this will include your full peer review and any attached files.

Reviewer #2: No

---

## [Editor Report · Acceptance letter]

12 Aug 2022

PONE-D-22-03571R2 

Contextualizing sacrificial dilemmas within Covid-19 for the study of moral judgment 

Dear Dr. Blanc:

I'm pleased to inform you that your manuscript has been deemed suitable for publication in PLOS ONE. Congratulations! Your manuscript is now with our production department. 

Kind regards, 

on behalf of

Dr. Sara Rubinelli 

Academic Editor

PLOS ONE